# Atherogenic Indices as a Predictor of Aortic Calcification in Prostate Cancer Patients Assessed Using ^18^F-Sodium Fluoride PET/CT

**DOI:** 10.3390/ijms232113056

**Published:** 2022-10-27

**Authors:** Michelle Dai, Winnie Xu, Helene Chesnais, Nancy Anabaraonye, James Parente, Shampa Chatterjee, Chamith S. Rajapakse

**Affiliations:** 1Department of Radiology, University of Pennsylvania, 1 Founders Building, 3400 Spruce Street, Philadelphia, PA 19104, USA; 2College of Osteopathic Medicine, Touro University Nevada, 874 American Pacific Drive, Henderson, NV 89014, USA; 3Department of Physiology, University of Pennsylvania, Philadelphia, PA 19104, USA; 4Department of Orthopaedic Surgery, University of Pennsylvania, Philadelphia, PA 19104, USA

**Keywords:** Sodium Fluoride, PET/CT, plaque, atherogenic indices, Castelli’s Risk Index, Atherogenic Index of Plasma, atherogenic coefficient, high-density lipoprotein cholesterol, vascular calcification, arteriosclerosis

## Abstract

A major pathophysiological cause of cardiovascular disease is vascular plaque calcification. Fluorine 18–Sodium Fluoride (18F-NaF) PET/CT can be used as a sensitive imaging modality for detection of vascular calcification. The aim of this study was to find a non-invasive, cost-efficient, and readily available metric for predicting vascular calcification severity. This retrospective study was performed on 36 participants who underwent 18F-NaF fused PET/CT scans. The mean standard uptake values (SUVs) were calculated from manually sectioned axial sections over the aortic arch and thoracic aorta. Correlation analyses were performed between SUVs and calculated atherogenic indices (AIs). Castelli’s Risk Index I (r = 0.63, *p* < 0.0001), Castelli’s Risk Index II (r = 0.64, *p* < 0.0001), Atherogenic Coefficient (r = 0.63, *p* < 0.0001), Atherogenic Index of Plasma (r = 0.51, *p* = 0.00152), and standalone high-density lipoprotein (HDL) cholesterol (r = −0.53, *p* = 0.000786) were associated with aortic calcification. AIs show strong association with aortic arch and thoracic aorta calcifications. AIs are better predictors of vascular calcification compared to standalone lipid metrics, with the exception of HDL cholesterol. Clinical application of AIs provides a holistic metric beneficial for enhancing screening and treatment protocols.

## 1. Introduction

Cardiovascular disease (CVD) is the leading cause of death in the United States [1]. Recent studies have shown that arteriosclerosis and hypercholesterolemia involve underlying oxidative stress mechanisms that lead to endothelial dysfunction [2,3,4]. Hypercholesterolemia induces activation of a proinflammatory cascade that damages vascular endothelium, leading to atherosclerotic plaque formation [5].

High-density lipoprotein (HDL) cholesterol and low-density lipoprotein (LDL) cholesterol play major roles in plaque formation and changes in plaque morphology [6]. These plaques are subject to inflammatory processes that develop vascular calcifications [7]. Vascular plaque instability can lead to rupture, which poses catastrophic health risks for ischemic cardiovascular complications, including myocardial infarction and stroke [8]. Therefore, early detection of unstable plaque is paramount in improving long-term patient health outcomes and decreasing overall costs of medical care.

Previous studies have shown that although the diagnosis of hypercholesterolemia can help identify individuals at risk of developing atherosclerotic diseases, it alone is insufficient for demonstrating the instability and thrombogenic risk of vascular plaques in detail [5].

Quantification of vascular calcification is currently regarded as the best single test for risk assessment in cardiovascular (CV) events. Therefore, a quantification of the relationship between lipid metrics and vascular plaque calcification will aid the development of accurate risk prediction from atherogenic indices. Here, we investigate standalone lipid panel values and atherogenic indices for potential use as clinical indicators of vascular calcification extent, and consequently, a measure of cardiovascular risk.

Fluorine 18–Sodium Fluoride (18F-NaF) PET/CT imaging incorporates radiolabeled 18F into areas of calcium deposition by exchanging fluorine with hydroxyl ions of hydroxyapatite crystals, forming fluorapatite [9,10]. This technique has shown remarkable specificity and sensitivity for early detection of vascular calcifications [5] that would otherwise go undetected on CT imaging alone [11,12]. 18F-NaF PET/CT has demonstrated a myriad of uses, ranging from cardiovascular conditions such as coronary and carotid atherosclerosis [13,14,15], to the evaluation of metastatic bone diseases, autoimmune diseases, and osteogenic bone disorders [16,17]. However, the current availability of 18F-NaF PET/CT imaging remains limited as a consequence of its invasive, costly, and time-consuming nature.

The goal of this study was to identify cost and resource-efficient, longitudinally available, and non-invasive metrics that can serve as clinical indicators for the extent of vascular calcification present in individuals. These metrics’ predictive powers can assist in the early clinical detection of cardiovascular diseases, as well as improve existing screening and risk stratification protocols for susceptible individuals.

## 2. Results

The atherogenic indices of CRI-I, (Figure 1), CRI-II (Figure 2), and AC (Figure 3) showed strong, positive correlations with mean aortic 18F-NaF tracer uptake. AIP (Figure 4) showed a moderate, positive correlation with mean aortic tracer uptake. Results are summarized in Table 1.

Among individual lipid parameters, HDL alone was significant for a moderate, negative correlation with mean aortic 18F-NaF tracer uptake (Table 2). These results remained significant even after controlling for age, weight, height, BMI, usage of statin therapy, and tracer dose.

## 3. Materials and Methods

This retrospective study was performed using clinical data obtained from the University of Pennsylvania’s healthcare system. Patients receiving 18F-NaF PET/CT imaging for evaluation of prostate cancer metastasis were selected based on PET/CT scan availability. The Institutional Review Board approved the study protocol, and the study was exempted from continuing review. All data was de-identified; therefore, written consent did not need to be obtained from patients.

### 3.1. Participant Selection

Criteria for inclusion in this study were male sex, availability of 18F-NaF PET/CT scans of the thorax for evaluation of previous diagnosis of prostate cancer, and lipid panel results performed within 365 days of the corresponding 18F-NaF PET/CT scan used in this study. 36 total participants from the University of Pennsylvania’s healthcare system met the inclusion criteria for this study.

### 3.2. 18F-NaF PET / CT

Practice guidelines recommend a tracer dose between 5–10 mCi for adults undergoing 18F-NaF PET/CT scans (Philips Healthcare, Cleveland, OH, USA), administered intravenously [18]. The mean tracer dose for the 36 participants was 8.11 ± 2.60 mCi F-18 NaF, and the mean time from tracer administration to scan performance was 68.12 ± 13.10 min. 18F-NaF PET/CT scans and radiology reports were obtained from radiological records.

### 3.3. Imaging Analysis

18F-NaF PET/CT scans were viewed with an imaging processor (Fiji PET/CT Viewer), allowing molecular information from PET scans to be overlaid with corresponding anatomical information from CT scans. To quantify the amount of 18F-NaF tracer uptake in the aorta as a measure of vascular calcification, regions of interest (ROIs) were manually selected encompassing the aorta in 18F-NaF PET/CT scans. Planar ROIs were sectioned per individual axial slice, starting superiorly from the level of the aortic arch, progressing inferiorly to the end of the thoracic aorta (Figure 5).

Immense care was taken to exclude possible sources of SUV contamination, including portions of the aortic arch bordering the sternum and thoracic spine, as well as portions of the thoracic aorta in proximity to the thoracic spine. Radiology reports were also referenced to exclude regions of notable calcification along the axial skeleton resulting from metastasis of prostate cancer to bony structures. All exclusions were reviewed together amongst all data collectors, and decisions to exclude individual portions of ROIs were made judiciously with all researchers present.

The mean standard uptake values (SUVmean) of 18F-NaF tracer were calculated for each axial slice via Fiji. Per-slice SUVmean values were then averaged along the total volume of the aortic arch and thoracic aorta.

### 3.4. Individual Lipid Metrics and Atherogenic Indices

Lipid profile measurements were accessed through electronic medical records. A total of 36 participants had lipid panel results dated within 365 days of the corresponding 18F-NaF PET/CT scan date, and were included in this study.

Serum concentrations in units of milligrams per deciliter (mg/dL) were recorded for high density lipoprotein (HDL) cholesterol, low density lipoprotein (LDL) cholesterol, total cholesterol (TC), and triglyceride (TG) levels. One participant had triglyceride levels too high (≥400 mg/dL) for an accurate LDL calculation as per the Friedewald equation [19], and was subsequently excluded from any statistical analyses involving LDL or atherogenic indices computed using LDL.

The derived values of non-HDL cholesterol [20] and atherogenic indices [21] investigated in this study were computed as shown in Table 3.

### 3.5. Statistical Analysis

Statistical analyses were performed in JMP version 16.0 (JMP, SAS Institute Inc., Cary, NC, USA). Statistical significance was set to *α* = 0.05. Pearson’s correlation analyses were performed on individual and derived lipid metrics against mean aortic SUV values as a measure of aortic calcification.

To correct for the potential effect of confounding factors such as age, weight, height, BMI, usage of statin therapy, and administered 18F-NaF tracer dose on SUVmean, multiple regression analyses were performed in JMP version 16.0.

## 4. Discussion

### 4.1. Pathophysiology of Plaque Calcification

Atherosclerotic plaque formation begins with vessel wall damage, allowing LDL to seep into the intimal layer. LDL oxidized (ox-LDL) by free radicals is phagocytosed by macrophages, forming foam cells and producing highly thrombogenic lipid cores encased in fibrous capsules. Throughout this highly inflammatory process, vascular smooth muscle cells (VSMCs) undergo osteogenic differentiation, creating microcalcifications [5,23,24].

HDL cholesterol’s role in plaque development is diametrically opposed to that of LDL cholesterol. HDL participates in reverse cholesterol transport (RCT), fetching excess cholesterol from peripheral tissues and returning it to the liver [23]. Its efficacy in cholesterol efflux from vessel walls hampers the pace of atherosclerotic plaque and calcification formation, and can even promote plaque regression [25,26,27].

In the midst of lipoprotein transport and metabolism, TG plays a regulatory role. Although it is not significant independently, TG has been shown to affect interactions between LDL and HDL cholesterol [28,29]. Elevated levels of TGs coupled with endothelial dysfunction initiates fatty streak formation and paves the way for plaque calcification [30].

### 4.2. Association between Atherogenic Indices and Arterial Calcification

The results of this study provide compelling evidence that atherogenic indices are strongly associated with the extent of aortic calcification. Atherogenic indices and standalone HDL cholesterol values exhibit greater strength of association than any other independent lipid parameters (TC, LDL, Non-HDL, TG) [21].

Among the atherogenic indices, CRI-I, CRI-II, and AC were strongly associated with aortic calcification amount. The similarity in correlation coefficients can be elucidated by the mathematical relationship between TC, LDL, and non-HDL cholesterol: about two-thirds of total plasma cholesterol is found in LDL [31], and the values of non-HDL cholesterol can be calculated by subtracting HDL from TC [20]. By nature of its derivation, CRI-II isolates LDL and HDL cholesterol levels, which are most directly involved in plaque formation and regression. The comparable association strengths among CRI-I, CRI-II, and AC allows for the substitution of CRI-I or AC in place of CRI-II when the value of LDL is unavailable; namely, when TG ≥ 400 mg/dL [19]. AIP encapsulates the modulatory effect of TG on HDL cholesterol, conveying more information about plasma atherogenicity contributors than TG independently [21].

Standalone HDL cholesterol is significant for a moderate, negative correlation with aortic calcification, demonstrative of its protective and anti-inflammatory properties [26]. However, although standalone HDL was inversely proportional to the severity of vascular calcification compared to other independent lipid parameters (TC, Non-HDL, LDL, TG), atherogenic indices CRI-I, CRI-II, and AC better encapsulates the complex interplay amongst these parameters in plaque formation. A side-by-side comparison between two study participants controlled for age, weight, height, BMI, and tracer dose demonstrates clear differences in calcification of the thoracic aorta (Figure 6), atherogenic indices, and HDL cholesterol levels (Table 4).

The results of this study support current research in evaluating the clinical use of lipid ratios over individual lipid metrics. Momiyama et al. evaluated the severity of thoracic aortic atherosclerosis via MRI over 95 participants, and found the CRI-II ratio to be more strongly associated with thoracic aortic atherosclerosis than HDL or LDL levels alone [32]. A cross-sectional study over 340 individuals by Fernández-Macías et al. further supported the clinical use of AIP as a biomarker for cardiovascular screening in developing countries [33]. A large study of 2676 adults performed by Onat et al. demonstrated that AIP showed the strongest association with hypertension, diabetes, and cardiovascular disease [34].

Studies on atherogenic indices in relationship to vascular calcification at sites other than the aortic arch and thoracic aorta have also been performed. Yildiz et al. performed mammogram analyses on 60 pre-menopausal women, and found a significant positive correlation between breast arterial calcification and and AIP, AC, CRI-I and CRI-II [35]. A study on 120 patients by Elsadek Seaoud et al. experimented with finding cut-off values for CRI-I and CRI-II that optimized sensitivity and specificity of these lipid ratios for coronary artery calcification [36].

Atherogenic indices have been shown to have clinical potential. Further research remains to be done on putting lipid ratios into application: setting diagnostic cutoff values for disease screening, normal ranges for therapeutic goals, and exploring other related sites of plaque formation and calcification.

### 4.3. 18F-NaF PET/CT Imaging in Vascular Calcification

18F-NaF’s mechanism of action involves chemisorption onto hydroxyapatite. Consequently, the surface area of the plaque and its calcification activity has an effect on tracer uptake. In a study performed on 75 individuals by Derlin et al., sites of 18F-NaF tracer accumulation on PET scans showed high colocalization (88%) with visible calcification sites on CT scans. However, the reverse did not hold true: there were far fewer (12%) sites of visible calcification on CT that showed corresponding colocalization with 18F-NaF tracer uptake on PET scans [37]. This suggests that some sites of microcalcification may be undetectable on CT scans, and that stable macrocalcifications may have low levels of 18F-NaF uptake due to low amounts of active calcium deposition. A study by Irkle et al. sought to understand this signal mismatch using ex vivo imaging studies on carotid endarterectomy sections. Their results were suggestive of plaque surface area playing a significant role in tracer adsorption. Macrocalcifications with large volume but small surface area showed proportionally less 18F-NaF uptake than microcalcifications with small volume, but a large surface area [11]. Because of its ability to detect early stage microcalcifications, incorporation of 18F-NaF PET/CT scanning has been recommended as part of routine coronary artery disease assessment [38].

The implications of these findings on this study are that the collected aortic SUVmean values may yield information about active calcium deposition in microcalcifications not yet visible on CT scans, but could underestimate the total amount of calcification present secondary to lack of active calcium deposition in stable macrocalcifications. Further research in this area may elucidate novel methods in estimating plaque volume based on calcification surface area.

### 4.4. Study Limitations

Several limitations to this study should be mentioned. The study cohort was comprised of male participants receiving 18F-NaF PET/CT scans for evaluation of prostate cancer. Although these results agree with findings about atherogenic indices from other populations [32,33,39], limitations remain on the generalizability of these findings. Furthermore, while other studies have found significant relationships between TC, LDL, Non-HDL, TG and vascular calcification [40], the discriminatory power in this study is limited by small cohort size (n = 35, 36). Some study participants did not have available LDL cholesterol values as a result of TG concentrations exceeding 400 mg/dL [19], and were therefore excluded. While the results of this study are promising, further investigation is warranted with a larger sample of diverse individuals.

## 5. Conclusions

Clinical use of atherogenic indices would provide clinicians with a more holistic understanding of plaque formation over standalone lipid panel metrics. Independent lipid parameters, while significant in the case of HDL cholesterol alone, do not provide information on the dysregulation of lipid metabolism and resulting calcification compared to atherogenic indices. The inclusion of the CRI-I, CRI-II, AC and AIP as part of standard lipid panel results is inexpensive, non-invasive, and readily available. Having such metrics in widespread use paves the way for improving current screening and risk assessment protocols for cardiovascular diseases. With CVD as the leading cause of death in the United States [1], the prevention, diagnosis, monitoring, and treatment of illnesses involving vascular calcification is paramount to improving national health outcomes.

## Figures and Tables

**Figure 1 ijms-23-13056-f001:**
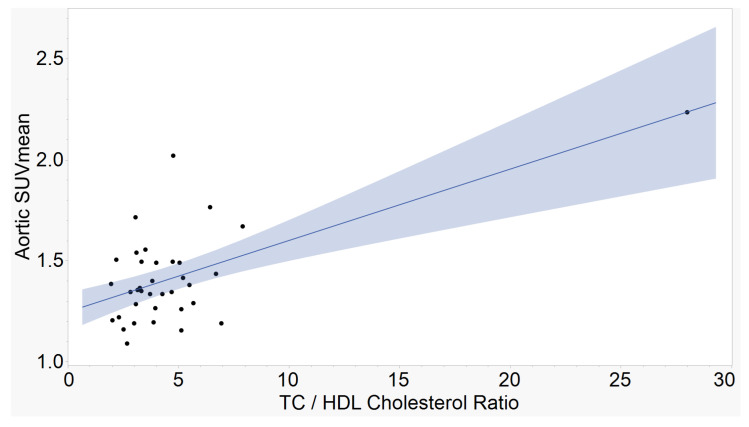
Aortic SUVmean vs. Castelli’s Risk Index I (CRI-I). The ratio between total cholesterol and high-density lipoprotein cholesterol demonstrates a strong, positive correlation with mean tracer uptake in the aorta. (r = 0.63, *p* < 0.0001, n = 36).

**Figure 2 ijms-23-13056-f002:**
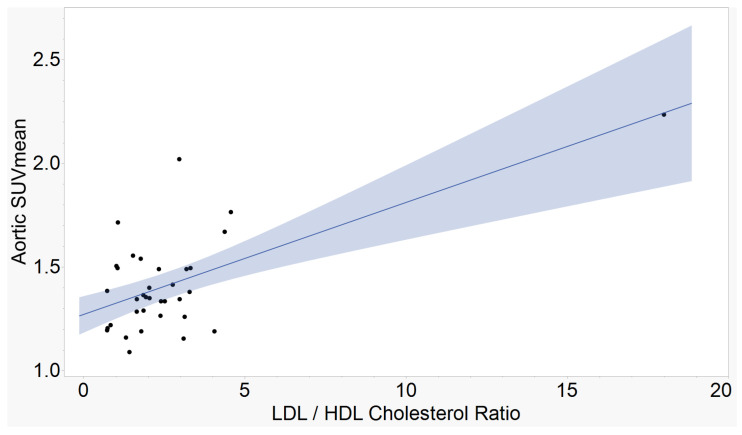
Aortic SUVmean vs. Castelli’s Risk Index II (CRI-II). The ratio between low-density lipoprotein and high-density lipoprotein cholesterol demonstrates a strong, positive correlation with mean tracer uptake in the aorta. (r = 0.64, *p* < 0.0001, n = 35).

**Figure 3 ijms-23-13056-f003:**
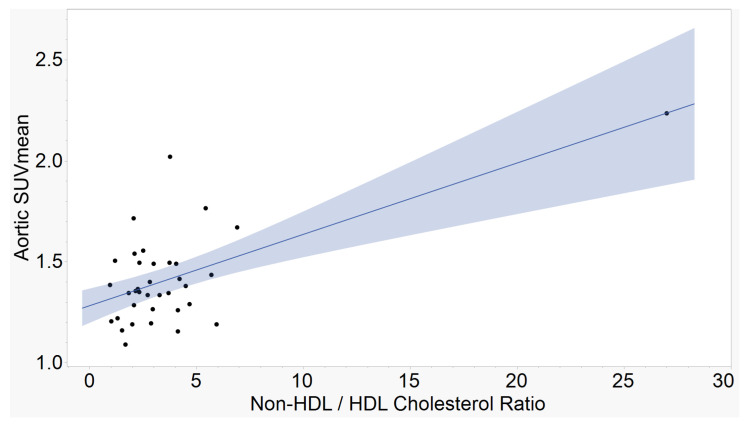
Aortic SUVmean vs. Atherogenic Coefficient (AC). The ratio between non-high density lipoprotein cholesterol and high-density lipoprotein cholesterol demonstrates a strong, positive correlation with mean tracer uptake in the aorta. (r = 0.63, *p* < 0.0001, n = 36).

**Figure 4 ijms-23-13056-f004:**
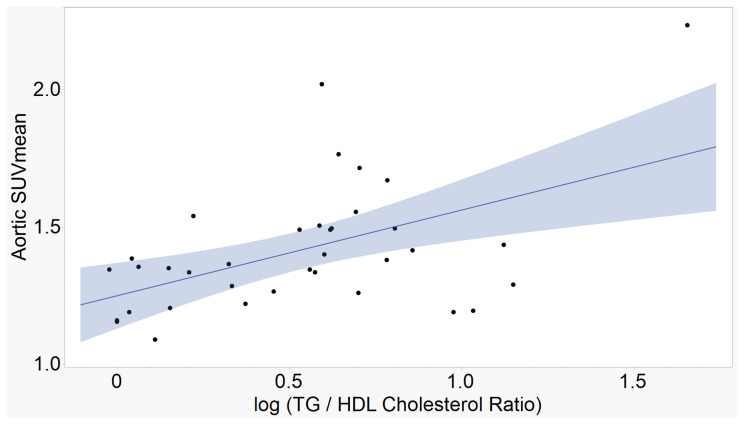
Aortic SUVmean vs. Atherogenic Index of Plasma (AIP). Calculated as log(triglycerides / HDL cholesterol), the AIP demonstrates a moderate, positive correlation with mean tracer uptake in the aorta. (r = 0.51, *p* = 0.00152, n = 36).

**Figure 5 ijms-23-13056-f005:**
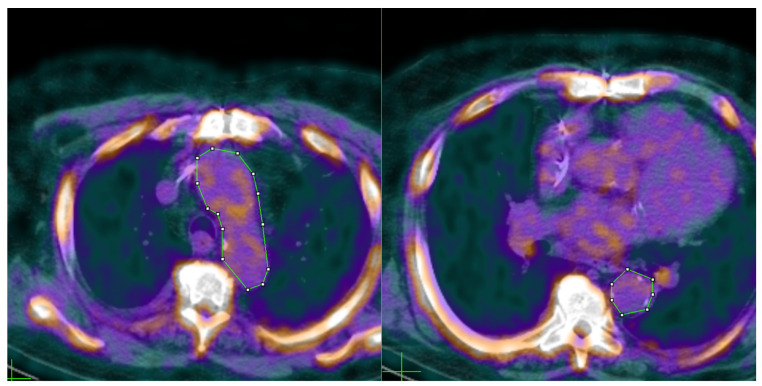
Fused PET/CT images showing example segmentations of the aorta. ROIs were manually selected on axial slices of the thorax. Colored heat maps of the PET scan served as a visual indicator of 18F-NaF tracer uptake, which was overlaid on structural anatomical information from the CT scan. For each ROI, SUVmean was recorded in Fiji Image Viewer. **Left**: A manually selected ROI outlined in green is shown encompassing the aortic arch. **Right**: An ROI taken from the thoracic aorta.

**Figure 6 ijms-23-13056-f006:**
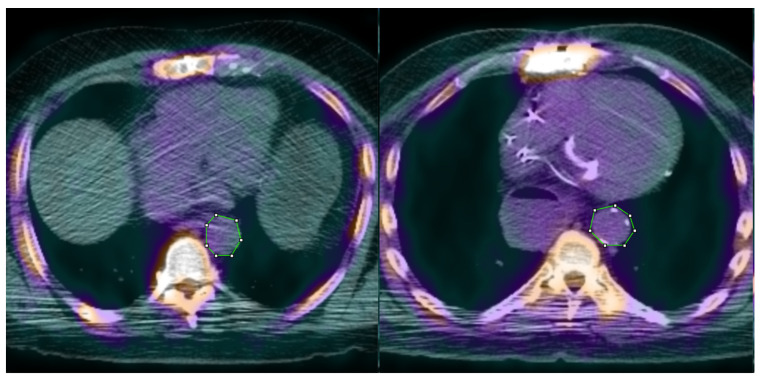
Visual comparison of axial view thoracic aorta 18F-NaF fused PET/CT scans between two study participants controlled for age, weight, height, BMI, and tracer dose. **Left**: study participant with fewer visible calcifications on CT and lower SUV intensity. **Right**: study participant with clearly visible aortic calcifications on CT scan and higher SUV intensity.

**Table 1 ijms-23-13056-t001:** Correlations between atherogenic indices and mean aortic tracer uptake.

Atherogenic Index	Correlation	Significance
CRI-I	0.63	<0.0001
CRI-II	0.64	<0.0001
AC	0.63	<0.0001
AIP	0.51	0.00152

**Table 2 ijms-23-13056-t002:** Correlations between individual lipid parameters and mean aortic tracer uptake.

Individual Parameter (mg/dL)	Correlation	Significance
HDL Cholesterol	−0.53	0.000786
Non-HDL Cholesterol	0.23	0.167
LDL Cholesterol	0.19	0.273
Triglycerides	0.19	0.269
Total Cholesterol	0.09	0.601

**Table 3 ijms-23-13056-t003:** Formulae of selected atherogenic indices and lipid metrics.

Lipid Metric	Formula
Non-HDL Cholesterol (Non-HDL)	TC − HDL cholesterol
Castelli’s Risk Index I (CRI-I)	TC/HDL cholesterol [22]
Castelli’s Risk Index II (CRI-II)	LDL/HDL cholesterol [22]
Atherogenic Coefficient (AC)	Non-HDL/HDL cholesterol
Atherogenic Index of Plasma (AIP)	log (TG/HDL cholesterol)

**Table 4 ijms-23-13056-t004:** Lipid metric comparison between two study participants shown above in Figure 6. **Left**: study participant with lower atherogenic indices and higher HDL cholesterol. **Right**: study participant with higher atherogenic indices and lower HDL cholesterol.

	Left	Right
Age (years)	68	71
Weight (kg)	100.2	99.3
Height (m)	1.83	1.77
BMI (kg/m^2^)	29.97	31.42
18F-NaF dose (mCi)	9.8	10.4
CRI-I	3.08	4.74
CRI-II	1.65	3.31
AC	2.08	3.74
AIP	0.33	0.62
HDL (mg/dL)	51	35
Aortic SUVmean	1.285	1.495

## Data Availability

Not applicable.

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
