# Peer review of "Atherogenic Indices as a Predictor of Aortic Calcification in Prostate Cancer Patients Assessed Using 18F-Sodium Fluoride PET/CT"

_ijms, 2022, doi:10.3390/ijms232113056_

Round 1

Reviewer 1 Report

Computed tomography, which detects macroscopic vascular deposits of calcium, has until recently been the only available noninvasive imaging modality  allowing visualization of this process.  The calcification activity now can be observed  using 18F-sodium fluoride positron emission tomography (PET) in combination with either computed tomography or magnetic  resonance. 18F-sodium fluoride positron emission tomography can be used to study microcalcification activity across the vasculature. 

The results of this study provide compelling evidence that atherogenic indices  are strongly  associated with the extent of aortic calcification. The atherogenic indices  showed  strong, positive, correlations with mean aortic 18F-NaF tracer uptake. The inclusion of the CRI-I, CRI-II, AC and AIP as part of standard lipid panel  results is inexpensive, non-invasive and readily  available. These results remained significant even after controlling for age, weight, height, BMI, usage of statin therapy and tracer dose.  Atherogenic indices have been shown to have potential  in clinical application.  A particular aspect was the fact that incorporation the 18F-NaF  tracer during  PET/CT scanning might prove beneficial for assessing  early-stage plaque calcification in coronary arteries disease. The bibliography presented includes recent titles. The article is well written and documented. I recommend it for publication. 

Reviewer 2 Report

The reviewed publication concerns mayby not the newest topic, but very important in clinical routine. Complications of atherosclerosis are very common and constitute a significant cause of death. the proposed and described method for assessing the activity of atherosclerotic plaques is made using an accessible and simple technique that has been shown to correlate with laboratory parameters and can be used in daily practice

The publication is prepared in a well-structured manner, material and methods are carefully and thoughtfully selected, and is very accessible to read. It seems interesting to readers.

The selection and scope of literature is substantive and adequate to the topic.

The drawback of the study is a small study group, which the authors of the publication are aware of and indicate the reason for this, suggesting the need to extend the study. However, this does not detract from the value of the work as a preliminary report.

Besides, I have no negative comments and I congratulate the authors of the work.